# Effect of Eggplant Powder on the Physicochemical and Sensory Characteristics of Reduced-Fat Pork Sausages

**DOI:** 10.3390/foods10040743

**Published:** 2021-04-01

**Authors:** Yuxia Zhu, Yawei Zhang, Zengqi Peng

**Affiliations:** 1National Center of Meat Quality and Safety Control, College of Food Science and Technology, Nanjing Agricultural University, Nanjing 210095, China; 2016208019@njau.edu.cn (Y.Z.); 2015009@njau.edu.cn (Y.Z.); 2Synergetic Innovation Center of Food Safety and Nutrition, Nanjing 210095, China

**Keywords:** eggplant powder, low-fat sausages, physicochemical characteristics, sensory properties

## Abstract

In this study, we investigated the effects of soybean oil, water, and 1, 2, and 3% eggplant powder (EP) as substitutes for pork back fat (a decrease from 30% to 15%) on the proximate composition, water- and fat-binding properties, colour, water distribution, texture, and sensory properties of pork sausages. The replacement of fat with soybean oil in sausages decreased the fat and cholesterol proportions and increased the moisture content, but the water- and fat-binding properties, texture properties, and sensory properties became worse. By adding EP, sausages displayed remarkably better water- and fat-binding properties, texture properties, and sensory properties. Moreover, EP addition significantly accelerated T_2_ relaxation time, increased the immobilised water content, and decreased the free water content of sausages. Sausages with 2% EP had the highest flavour, mouthfeel, and overall acceptability. This work provides theoretical and data support for the preparation of sausages with EP rich in unsaturated fatty acids (UFAs) and dietary fibre.

## 1. Introduction

Pork sausages are popular, frequently consumed meat products that occupy a large proportion of the meat market for their smooth taste, fine texture, nutrition-rich properties, and convenient use [1]. Commercial sausages contain approximately 30% fat, which contributes to the characteristics of the quality of meat products, such as flavour, texture, juiciness, colour, and mouthfeel, thus enhancing the acceptability of these products [2]. However, high animal fat content in meat products increases the incidence of diet-related diseases, such as obesity, hypertension, type-2 diabetes, cardiovascular diseases, and coronary heart diseases, owing to high amounts of saturated fatty acids (SFAs) and serum cholesterol [3]. The meat industry is entering a new era where products are characterised by low fat, low calorie, and high fibre [4]. Such characteristics are driven by consumer demands for healthier and more nutritious diets, leading to the development of innovative products with reduced fat in traditional formulations.

In recent years, a variety of plant oils [2,5,6,7,8,9,10], such as olive oil, sesame oil, sunflower oil, corn oil, canola oil, grape seed oil, hazelnut oil, and soybean oil have been added to various meat products as partial substitutes for animal fats, which can reduce the proportion of SFAs and increase the proportion of unsaturated fatty acids (UFAs). Among vegetable oil sources, soybean is widely planted worldwide with a high yield. Therefore, soybean oil is a low-cost and easily obtained vegetable oil that contains a high proportion of UFAs (52–56% linoleic acid, 20–25% oleic acid, and 7–11% linolenic acid) [11]. However, vegetable oil added solely to meat batter may affect its appearance, flavour, texture, and physical stability, significantly decreasing the acceptability of meat products [2,12]. Consequently, emulsifiers and/or stabilisers that function as meat binders and texture stabilisers have been added to different meat products to counter the problem caused by fat substitutes, to reduce cooking loss, and to improve physicochemical and sensory properties [13,14].

Polysaccharides, including dietary fibre [5,15,16,17,18,19] (such as sugarcane, pumpkin, pineapple, rye bran, mushrooms, and rice bran), konjac [20], carrageenan [21], guar gum [22], regenerated cellulose [6], xylooligosaccharide [23], carboxymethyl cellulose, and microcrystalline cellulose [24], are widely used candidates of importance. Moreover, dietary fibre derived from plant polysaccharides is widely accepted as an important nutrient in the human diet, based on its ability to improve intestinal functions, decrease the incidence rates of cardiovascular and gastrointestinal diseases, and have a protective effect against weight gain and obesity without reducing the feeling of satiety [4,14,16,25].

Eggplant (*Solanum melongena* L.) is widely planted worldwide [26], and is reported to have a rich source of essential nutrients, with a high content of dietary fibre (33.79% in dry weight) and protein (11.65%), and low level of fat [27]. Its incorporation in meat products may improve the dietary fibre content of meat products, thus providing well-balanced meals for meat eaters. In our previous study [27], it was found that the mechanically homogenised eggplant flesh pulp (EFP) can effectively emulsify soybean oil in water emulsions (3:7, *v*/*v*) with good viscoelasticity by forming an interfacial film adhered to the oil droplets and the coherent three-dimensional network in the continuous phase. Rheological analysis illustrated that the EFP emulsion had typical gel-like nature. Moreover, the whole eggplant without chemical treatment can be utilised as food supplement, which leads to an environmentally friendly way for food production. Considering the features of eggplant, it could be a promising naturally functional ingredient to improve the physicochemical and sensory properties of fat-reduced meat products. In addition, the use of eggplant as an additive in meat products has not been previously investigated.

As a contribution to the development of handier and more versatile procedures for replacing animal fat with soybean oil, we conducted a study to assess the effect of 1, 2 and 3% eggplant powder (EP) addition on physicochemical properties (proximate composition, fat and water-binding properties, colour, water distribution, and texture) and sensory characteristics of reduced-fat pork sausages. The results of this study may provide a basis for studies on the preparation of EP sausages rich in UFAs and dietary fibre. 

## 2. Materials and Methods

### 2.1. Materials

Pork leg muscle (71.82% moisture, 3.26% lipids, and 23.64% protein) and pork back fat (8.62% moisture, 89.64% lipids, and 2.11% protein) were purchased from Yurun chilled meat (Nanjing, China); purple eggplant fruits (*Solanum melongena* L.) with accordant maturity (length 20 cm and diameter 6 cm), soybean oil, salt, sugar, and white pepper from local supermarket (Nanjing, China); sodium caseinate (protein > 90%) from Henan Wanbang Industrial (Zhengzhou, China); sodium tripolyphosphate (food grade) from Nanjing Zhonghua Seasoning (Nanjing, China); and commercial cholesterol analysis test kit (F002-1, Zhejiang Dongou, Wenzhou, China) from Nanjing Jiancheng Bioengineering Institute (Nanjing, China).

### 2.2. Preparation of Eggplant Powder

The peeled eggplant was sliced and dried in a fan-forced air oven (DGG-9240A, Senxin Company, Guangzhou, China) at 60 °C for 36 h. Then, the dried eggplant was ground in a centrifugal mill (TLG-08, Tianlihengcheng, Beijing, China) fitted with a 30-mesh screen. The EP was stored in a vacuum pressure desiccator at room temperature until use.

### 2.3. Preparation of Pork Sausages

The pork leg muscle (all visible connective tissue and fat were trimmed) and back fat were cut into pieces and separately passed through a mincer (MM-12, Fengwei Product, Zhengzhou, China) using a 6 mm plate. The formulations of pork sausages are listed in Table 1. Briefly, the minced meat was chopped in a bowl chopper (UMC-5C, Stephan, Karlsruhe, Germany) at a low speed (1500 rpm) for 30 s, followed by the addition of salt, sodium tripolyphosphate, pepper, and sugar, and chopped at a high speed (3000 rpm) for 60 s. After a 120 s pause, back fat, soybean oil, sodium caseinate, EP, and 1/2 ice water were added and the mixture chopped for 120 s at high speed, followed by the addition of the remaining ice water and chopping for another 120 s after a 60 s pause. The temperature of the meat batter was kept below 12 °C in all samples.

A portion of ground meat was drawn (about 20 g sample) to analyse its water- and fat-binding properties, and the rest was used to prepare sausages. The meat batter was stuffed into nylon/PE casings with a diameter of 26 mm, manually linked into approximately 15 cm lengths, and heated in a water bath (HH-42, Guohua, Changzhou, China) at 80 °C for 20 min. The sausages were cooled at room temperature and stored in a freezer at 4 °C until subsequent analysis within a week of production.

### 2.4. Proximate Composition of the Pork Sausages

Moisture content was determined by weight loss after drying at 105 °C for 12 h in a drying oven (DGG-9240A, Senxin, Guangzhou, China) according to AOAC 950.46 [28]. Fat content was determined using the Soxhlet method (SOX406, Hanon, Jinan, China) according to AOAC 960.39 [28]. Protein content was determined using the Kjeldahl method (Kjeltec 2300, Foss, Beijing, China) according to AOAC 981.10 [28]. The ash content was determined using a muffle furnace (MF0910Pa, Huagangtong Technology, Beijing, China) according to AOAC 920.153 [28]. The cholesterol content was determined using a commercial cholesterol analysis test kit.

### 2.5. Water- and Oil-Binding Properties

Water- and oil-binding properties were determined by measuring water and oil loss according to a previously described procedure [5]. Raw batter (~20 g) was weighed (M_1_), transferred to 50 mL centrifuge tubes, and centrifuged at 3000 rpm for 15 min at 4 °C to remove air bubbles (Allegra 64R, Beckman Coulter, California, USA). The tubes were heated in an 80 °C water bath (HH-42, Guohua, Changzhou, China) for 20 min and then immediately uncapped and left inverted for 1 h to release fat and water exudate (total fluid release, TR) onto a weighing bottle (M_2_). Water release (WR) was determined as weight loss after heating the total exudate in an air oven (DGG-9240A, Senxin, Guangzhou, China) at 105 °C for 16 h (M_3_). Oil release (OR, ignored minor protein or salt component) was calculated as the difference between TR and WR:TR = (M_2_ − M_0_)/M_1_ × 100%(1)
WR = (M_2_ − M_3_)/M_1_ × 100%(2)
OR = (M_3_ − M_0_)/M_1_ × 100%(3)
where M_0_, M_1_, M_2_, and M_3_ are the weight of the weighing bottle, raw batter, total exudate together with the weighing bottle, and oil release together with the weighing bottle, respectively.

### 2.6. Colour Measurements

The colour of the sausages was measured using a colorimeter (CR-400, Minolta Camera, Tokyo, Japan) and Illuminate C, calibrated with a white plate (*L** = 96.86, *a** = −0.15, *b** = 1.87). Lightness (*L**), redness (*a**), and yellowness (*b**) values of the five measurements were recorded.

### 2.7. Low-Field Nuclear Magnetic Resonance (NMR) Relaxation Measurements

NMR relaxation measurement was performed on an NMR Analyzer (MesoMr23, Niumag, Suzhou, China) with the magnetic field strength of 0.5 Tesla, and with corresponding resonance frequency for protons of 21 MHz. Sausages (approximately 2.0 g) were placed into cylindrical glass tubes (15 mm in diameter) and transverse relaxation (T_2_) was measured at 32 °C using the Carr-Purcell-Meiboom-Gill pulse sequence. T_2_ measurements were performed with a τ-value (interval between 90° and 180° pulse width) of 140 µs, 6500 echoes as eight scan repetitions, and repetition time of 5 s. The obtained T_2_ data were subjected to multi-exponential fitting analysis using MultiExp Inv Analysis software (Niumag, Suzhou, China). The measurements were carried out with five replicates for each sample and expressed as T_2a_, T_2b_, T_21_, and T_22_.

### 2.8. Texture Profile Analysis (TPA)

The sausages were cut into five cylindrical segments (height 20 mm, diameter 26 mm) after equilibration to room temperature at 25 °C for 3 h. TPA was performed using a texture analyser (TA-XT2i, Stable Micro System, Surrey, UK) fitted with the loadcell of 50 kg and a cylindrical probe (P/50, 50 mm stainless cylinder). The TPA measurement conditions were, pre-test speed 2.0 mm/s, test speed 2.0 mm/s, post-test speed 5.0 mm/s, strain 50%, time 5.0 s, trigger type auto, and trigger force of 5 g [5]. All measurements were carried out at room temperature, and the hardness, springiness, cohesiveness, adhesiveness, and chewiness of the sausage were evaluated.

### 2.9. Sensory Analysis

Sensory evaluation was performed by 10 panellists who were trained according to the Chinese standard GB/T22210-2008 (criterion for sensory evaluation of meat and meat products) and had a common consensus for each point of the evaluation index. Sausages were warmed and served to the panellists for assessment of appearance, flavour, texture, mouthfeel, and overall acceptability using a 9-point hedonic scale [5,29]. The samples were cut into bite-sized pieces, placed on a plate, encoded with random numbers, and placed randomly. Sensory evaluation was performed at 2–3 h after the panellists’ meal, and the panellists had to gargle with distilled water before evaluating each sample. There was no communication among the panellists during the entire evaluation process. The sensory score was averaged after deducting the outliers (9, like extremely; 8, like very much; 7, like moderately; 6, like slightly; 5, neither like nor dislike; 4, dislike slightly; 3, dislike moderately; 2, dislike very much; 1, dislike extremely).

### 2.10. Statistical Analysis

Experiments were performed in triplicate (except for specially declared) for each sample. The data were analysed using the SAS v9.2 Windows program by an analysis of variance (one-way ANOVA) and Duncan’s multiple-range test. The results are reported as mean values ± standard deviations, and differences were considered to be significant when *p* < 0.05.

## 3. Results and Discussion

### 3.1. Proximate Composition of Pork Sausages

The moisture content of low-fat groups was significantly higher than that of the control (Table 2, *p* < 0.05), which resulted from the higher amount of additional water in the low-fat groups. The incorporation of EP further increased the moisture content of the low-fat sausages because EP addition can avert cooking loss owing to its strong water-holding capacity [27]. The fat content of the low-fat groups (R, EP1, EP2, and EP3) was significantly lower than that of the control (*p* < 0.05), but there was no significant difference among the treatments (*p* > 0.05). Therefore, the proposed formulation not only reduced the fat content, but also provided a way to increase the ratio of UFAs to SFAs, thereby satisfying the demands for low-fat meat products. These results are in agreement with a previous report [4], in which the replacement of pork back fat with sunflower oil in frankfurter significantly increased the moisture content and decreased the fat content. The protein content of sausages was approximately the same, ranging from 11.73% to 12.18%. The ash content of the sausages ranged from 1.86% to 2.02%, and it was slightly higher in the formulations containing EP than that in the control, which was attributed to the minerals in EP [30].

Furthermore, the low-fat sausages had significantly lower cholesterol contents (55.25 mg/100 g) than that in the control (71.90 mg/100 g) (*p* < 0.05). This was caused by the fact that fat was partly substituted by soybean oil in the low-fat sausages, where soybean oil is free of cholesterol and has a higher ratio of UFAs to SFAs than that in pork back fat [11]. A similar result was noted in Ref. [8], which reported that animal fat substituted by vegetable oil has a lower cholesterol content.

### 3.2. Water- and Oil-Binding Properties

Water- and oil-binding properties are important for determining the quality and yield of final meat products, as well as their subsequent texture and sensory properties. The TR, WR, and OR values of the sausages are shown in Table 3. The TR, WR, and OR of low-fat sausages without EP (R group) were significantly higher than those of the control (*p* < 0.05), whereas they were remarkably reduced with the addition of EP. Similar results have been reported in previous studies [1,5,15,19], which revealed that TR, WR, and OR can be improved by the addition of fibres to meat products. For low-fat sausages with different contents of EP, there was no significant difference of OR (*p* > 0.05). Considering the TR and WR, they were higher in the EP1 group than those in the control (*p* < 0.05). However, they were comparable to those of control when the added EP content was above 2% (*p* > 0.05). This is probably explained by the fact that a higher amount of water content resulted in higher exudation as the EP was insufficient (<2%) to bind the added water. In fact, eggplant polysaccharides are anionic charged polysaccharides that contain galacturonic acid (33.3%) [27]. For the meat batter with EP, the oil droplets and fat globules were surrounded by a thick dense layer of polysaccharide-protein electrostatic complexes, which led to an increase in steric repulsion and thus enhanced the stability of the meat emulsion system [31]. In this way, EP can bind the added water via hydrogen bonding and immobilise water in the gel network during gelling [32]. Therefore, the water- and oil-binding properties of meat products can be improved.

### 3.3. Colour Measurements

Colour is an important aspect of meat products because it influences the consumers to purchase the product. Replacement of pork back fat with soybean oil increased *L** and *b**-values and reduced *a**-values, as shown in Table 4. The *L**-value was highest in the low-fat sausage without EP addition (R group), whereas it significantly decreased with the addition of EP (*p* < 0.05). When the EP content was above 2%, the *L**-value was comparable to that of the control, which exhibited the lowest *L**-value (*p* > 0.05). The main reason for this variation is that evenly distributed emulsified vegetable oil with smaller droplet size has strong light reflectivity caused by optical scattering, whereas the fat globules with larger droplet sizes have weak light reflectivity owing to Rayleigh or Mie scattering [33]. The strong light reflectivity contributed to the increase in lightness (*L**-value). However, EP decreased the *L**-value of low-fat sausages, as the EP was brownish, which resulted from enzymatic browning caused by polyphenol oxidase during the drying process [34]. A similar phenomenon was reported in Ref. [35], in which the *L**-value of the frankfurter decreased significantly by increasing the pumpkin fibre because of its yellowish colour. The addition of EP further decreased the redness (*a**-value) of the sausages (*p* < 0.05), which was similar to the results reported by Choi [13], where the replacement of pork back fat by olive oil and dietary fibre reduced the redness of the frankfurter. The *b**-value of low-fat sausage without EP (R group) had no difference with that of the control (*p* > 0.05), whereas it significantly increased with the increase of EP levels (*p* < 0.05). Henning [16] found that meat products showed a higher *b**-value when yellow or white dietary fibre was added. Moreover, it was believed that the yellowish soybean oil also contributed to the higher *b**-value.

### 3.4. Low-Field Nuclear Magnetic Resonance (LF-NMR) Transverse (T_2_) Analysis

Distributed exponential fitting analysis of the T_2_ relaxation decay data was performed to investigate the qualitative and quantitative information regarding the physical state of water in meat products. Four distinct fractions of water populations were detected by LF-NMR and referred to as T_2a_, T_2b_, T_21_, and T_22_, respectively (Figure 1). A minor fraction between 0.1 and 1 ms (T_2a_) and another one between 1 and 10 ms (T_2b_) are associated with the bound water that is tightly bonded with fat and protein, respectively. A major fraction located between 10 and 100 ms represents the immobilised water, which reflects the portion of water trapped within the gel network structure. Finally, the free water fraction (T_22_), located between 100 and 1000 ms, represents the extra-gel water [36].

The relaxation times (T_2a_, T_2b_, T_21_, and T_22_) and corresponding proportions (P_2a_, P_2b_, P_21_, and P_22_) of the water molecule population for each relaxation component are given in Table 5. The representative distribution of T_2_ relaxation time reflects the water distribution and mobility in sausages, whereas the percentage of T_2_ relaxation area represents the relative fractions of water populations. The T_2a_ and T_2b_ relaxation times and their corresponding proportions P_2a_ and P_2b_ had no manifest differences for any of the treatments (*p* > 0.05), except for the P_2a_ proportion of the control, which was higher than that of the low-fat group (*p* < 0.05). T_2a_ and T_2b_ are the parts of moisture associated with fat and protein molecules through a covalent bond, and their variation depends on sausage formulation [37]. As observed in the results, the T_21_ relaxation time of the R group increased when compared to the control (*p* < 0.05), whereas it decreased with EP addition and with increasing contents of EP being associated with lower values (*p* < 0.05). For P_21_, it decreased (*p* < 0.05) in the R group when compared with the control, whereas it increased (*p* < 0.05) with the addition of EP. In contrast, the P_22_ proportion increased from 5.98% (control) to 8.94% (R group), whereas it decreased significantly in the low-fat group with EP addition. In conclusion, EP addition significantly accelerated the T_2_ relaxation time, increased the immobilised water content, and decreased the free water content, thereby significantly improving the loose gel network caused by the substitution of animal fat with water and vegetable oil. Similar results were achieved by Ran [36] using perilla seed and by Zhuang [38] using sugarcane insoluble dietary fibre as a sausage additive. This enhanced confinement of water is consistent with the improved water-binding properties of the low-fat sausages (Table 3). Similarly, Moller [39] reported that T_21_ is positively correlated with the water-holding capacity of meat products.

### 3.5. Texture Profile Analysis

Moreover, eggplant polysaccharides are anionic polysaccharides [27]. Hydrogen bonding and electrostatic interactions are the main molecular forces involved in the interaction of proteins and polysaccharides during gel formation [31]. The anion group of EP polysaccharide can bind water in the protein matrix through hydrogen bonding in the three-dimensional network structure of the gel.

Texture profile analysis (Table 6) indicated that reducing fat by partial replacement of animal fat with soybean oil and water (R group) presented lower hardness, adhesiveness, and chewiness values than those of the control. No significant differences in springiness and cohesiveness were observed among the treatments (*p* > 0.05). The decrease in textural properties can be attributed to an increase in water and reduction of fat [40]. Choi [35] also noted that a fat reduction strategy affects the textural properties of meat products. Cofrades [41] reported that frankfurters with high-fat were harder than low-fat ones, because the hardness of frankfurters is dependent on the amount of fat replaced by water. However, the hardness and chewiness values were increased through EP addition for the low-fat sausages compared to the R group without EP (*p* < 0.05), with increasing contents of EP being associated with higher values. The results are consistent with those reported by Zhuang [5], who noted that hardness, gumminess, and chewiness are higher when sugarcane dietary fibre is added to low-fat sausages, and by Kim [15], who reported that pumpkin fibre can significantly increase the hardness and chewiness of frankfurters.

Normally, dietary fibre has a strong water-holding ability and binding capacity, and can form an insoluble three-dimensional network to enhance the rheological properties of meat emulsion batter [42]. In general, the hardness of meat products is related to the viscoelastic properties and water-holding capacity of protein gels [40]. Eggplant was reported to be a good emulsifier because it is rich in fibre. The prepared EFP can thicken the stabilising layer around protein-coated droplets, form a three-dimensional network in the meat matrix, and improve the stability of meat emulsions and the textural properties of meat products [27,36]. The effects of dietary fibre on the improvement of texture properties differ from the source and nature of dietary fibre. For instance, Zhuang [5] reported that sugarcane dietary fibre can improve the texture of sausages by increasing salt-soluble protein (the major components of interface membrane) and immobilising water in meat batter. As a result, the stability of the meat batter system is greatly enhanced. Jiménez-Colmenero [2] noted that oil-in-water emulsion added to substitute part of pork fat in meat products can promote greater dispersion of the lipid phase with smaller particle size, and can increase the gel strength by the well-embedded small fat globules in the protein matrix. In addition, Do Amaral [43] found that chitosan can promote a firmer texture, which is ascribed to the fact that chitosan acts as a meat binder to form a stronger gel.

However, different results were reported by Park [44], who revealed that hardness is reduced by adding soluble chitosan into the pork gel, and by Choi [35], who reported that laminaria japonica decreases the hardness of low-fat meat products. Han [36] studied the textural properties of a fat-reduced model meat system enriched with inulin, cellulose, CMC, chitosan, and pectin, and found that CMC and pectin lead to a decrease and cellulose and chitosan to an increase in hardness, but inulin has no effect. The addition of these dietary fibres decreases the gel strength of meat products, probably because of the disruption of protein-water or protein-protein gel networks [24,44]. It is believed that the effect of dietary fibre on the hardness of meat products is related to its physicochemical properties, such as solubility, molecular weight, and hydrophobicity [45].

### 3.6. Sensory Analysis

The appearance, flavour, texture, mouthfeel, and overall acceptability of each sample were evaluated, and the values are shown in Figure 2. When compared to the control, the R group, in which soybean oil and water were simply used to replace fat, had a sharp decrease in appearance, flavour, texture, and overall acceptability scores (*p* < 0.05), indicating that a fat-reducing method by simply using water and vegetable oil to substitute animal fat was impracticable. However, the addition of EP to low-fat sausages increased the flavour, texture, mouthfeel, and overall acceptability scores, except for the appearance score (*p* < 0.05). In fact, partial substitution of animal fat with vegetable oil has a negative effect on appearance. In addition, the brownish colour of EP decreased the appearance score of the sausages. However, the addition of EP can reduce the greasiness of the sausages, and a richer and more desirable flavour was obtained because of its outstanding ability to absorb large amounts of fat [26]. Many researchers have reported the beneficial and positive effects of dietary fibre on the sensory properties of reduced-fat meat products, such as Zhuang [5], who reported that sugarcane dietary fibre significantly increases the texture and juiciness scores of reduced-fat meat sausages, and Choi [40], who noted that reduced-fat frankfurters with makgeolli lees fibre had higher tenderness, juiciness, and overall acceptability scores.

Sensory analysis showed that sausage formulated with 2% EP was the most acceptable to the panellists after considering appearance, flavour, texture, and mouthfeel. This is in contract to the research of Kehlet [46], who believed that meatballs with a rye bran fibre content of 3% could obtain a balance between sensory properties (odour, texture, and flavour) and nutrition, whereas a pea fibre content of 6% could obtain a more crumbly, firm, and gritty texture. In addition, Jiménez-Colmenero [2] reported that meatballs containing 4.60% dietary fibre have the most acceptable sensory characteristics by considering general appearance, taste, texture, juiciness, and general acceptability. The effects of different dietary fibre contents on the sensory properties of meat products might be related to the source of dietary fibre.

## 4. Conclusions

Reducing animal fat levels from 30% to 15% by adding soybean oil and water reduced the fat and cholesterol contents of sausages, but the water- and fat-binding properties, moisture stability, and texture were poor and the acceptability of sensory properties were reduced. To further improve these properties of reduced-fat sausages, various amounts of EP were added (1, 2, and 3%). With increasing EP contents, the sausages had increased water- and fat-binding properties, moisture stability, texture, and sensory properties. In particular, the low-fat sausages with 2% EP addition had the best acceptability in terms of appearance, flavour, texture, and mouthfeel. However, EP had a negative effect on the colour of the low-fat sausages, as it significantly decreased the *L**- and *a**-value. Moreover, the sensory evaluation was carried out by panellist instead of consumer. Overall, EP showed potential as a means of enhancing water- and fat-binding properties, moisture stability, texture, and sensory properties of sausages. As a future investigation, the mechanism underlying the physicochemical influence of EP on reduced-fat sausages would be assessed.

## Figures and Tables

**Figure 1 foods-10-00743-f001:**
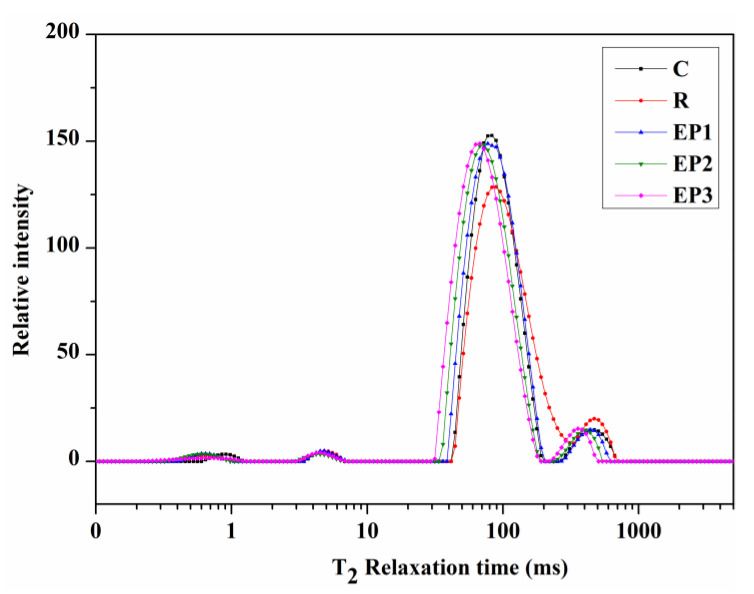
T_2_ relaxation time distribution of sausages.

**Figure 2 foods-10-00743-f002:**
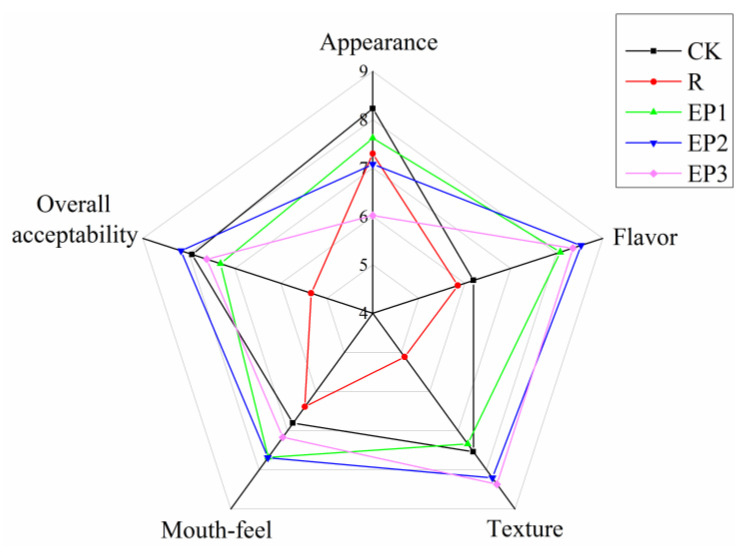
Sensory properties of pork sausages.

**Table 1 foods-10-00743-t001:** Formulation of pork sausages.

Ingredient	Treatment (g/100 g)				
	C	R	EP1	EP2	EP3
Pork leg muscle	50	50	50	50	50
Pork fat	30	15	15	15	15
Ice water	20	25	24	23	22
Soybean oil	0	10	10	10	10
EP	0	0	1	2	3
Total	100	100	100	100	100
Salt	1.5	1.5	1.5	1.5	1.5
Sodium caseinate	0.75	0.75	0.75	0.75	0.75
Sodium tripolyphosphate	0.2	0.2	0.2	0.2	0.2
White pepper	0.5	0.5	0.5	0.5	0.5
Sugar	0.5	0.5	0.5	0.5	0.5

Notes: C: 30% pork fat; R: 15% pork fat + 10% soybean oil; EP1: R + 1% EP; EP2: R + 2% EP; EP3: R + 3% EP.

**Table 2 foods-10-00743-t002:** Proximate composition of pork sausages.

Treatment	Cholesterol (mg/100 g)	Moisture (%)	Fat (%)	Protein (%)	Ash (%)
C	71.90 ± 1.12 ^a^	56.35 ± 0.55 ^c^	29.44 ± 0.68 ^a^	12.10 ± 0.13	1.86 ± 0.09
R	55.25 ± 1.65 ^b^	58.80 ± 0.50 ^b^	25.28 ± 0.78 ^b^	11.73 ± 0.14	1.89 ± 0.05
EP1	55.05 ± 1.85 ^b^	59.20 ± 0.30 ^b^	25.02 ± 0.34 ^b^	11.96 ± 0.12	1.91 ± 0.15
EP2	56.05 ± 1.15 ^b^	61.00 ± 0.50 ^a^	24.86 ± 0.74 ^b^	12.18 ± 0.12	1.94 ± 0.12
EP3	54.75 ± 1.55 ^b^	60.61 ± 0.25 ^a^	24.56 ± 0.11 ^b^	12.16 ± 0.18	1.92 ± 0.08
*F* Value	24.94 **	15.47 **	11.79 **	1.29	0.09

Notes: C: 30% pork fat; R: 15% pork fat + 10% soybean oil; EP1: R + 1% EP; EP2: R + 2% EP; EP3: R + 3% EP. The data in the table are expressed as means ± standard deviation. ^a–c^: Different letters within the same column mean significant differences (*p* < 0.05), no letters within the same column mean no significant differences (*p* > 0.05). Symbol represents *p* value (** *p* < 0.01).

**Table 3 foods-10-00743-t003:** Water-and fat-binding properties of pork sausages.

Treatment	TR (%)	WR (%)	OR (%)
C	2.52 ± 0.26 ^c^	2.22 ± 0.32 ^c^	0.29 ± 0.06 ^b^
R	8.63 ± 0.36 ^a^	6.38 ± 0.29 ^a^	2.25 ± 0.65 ^a^
EP1	5.64 ± 0.09 ^b^	4.77 ± 0.31 ^b^	0.87 ± 0.22 ^b^
EP2	3.21 ± 0.26 ^c^	2.86 ± 0.16 ^c^	0.35 ± 0.10 ^b^
EP3	2.72 ± 0.21 ^c^	2.52 ± 0.26 ^c^	0.20 ± 0.05 ^b^
*F* Value	66.68 **	41.76 **	7.52 *

Notes: C: 30% pork fat; R: 15% pork fat + 10% soybean oil; EP1: R + 1% EP; EP2: R + 2% EP; EP3: R + 3% EP. The data in the table are expressed as means ± standard deviation. ^a–c^: Different letters within the same column mean significant differences (*p* < 0.05). * Symbol represents *p* value (* 0.01 ≤ *p* < 0.05, ** *p* < 0.01).

**Table 4 foods-10-00743-t004:** Colour parameters (*L**, *a** and *b**-value) of pork sausages.

Treatment	*L**	*a**	*b**
C	75.58 ± 0.43 ^c^	5.49 ± 0.25 ^a^	10.70 ± 0.21 ^d^
R	80.24 ± 0.06 ^a^	3.52 ± 0.05 ^b^	10.92 ± 0.16 ^d^
EP1	77.13 ± 0.57 ^b^	2.94 ± 0.38 ^c^	12.32 ± 0.16 ^c^
EP2	76.27 ± 0.32 ^b,c^	2.23 ± 0.12 ^d^	15.57 ± 0.16 ^b^
EP3	75.72 ± 0.30 ^c^	2.25 ± 0.06 ^d^	16.20 ± 0.39 ^a^
*F* Value	32.18 **	52.03 **	80.50 **

Notes: C: 30% pork fat; R: 15% pork fat + 10% soybean oil; EP1: R + 1% EP; EP2: R + 2% EP; EP3: R + 3% EP. The data in the table are expressed as means ± standard deviation. ^a–d^: Different letters within the same column mean significant differences (*p* < 0.05). * Symbol represents *p* value (** *p* < 0.01).

**Table 5 foods-10-00743-t005:** T_2_ relaxation time and corresponding population of sausages.

Treatment	T_2a_	T_2b_	T_21_	T_22_	P_2a_	P_2b_	P_21_	P_22_
C	0.81 ± 0.06	4.61 ± 0.29	85.92 ± 2.08 ^b^	462.30 ± 39.31 ^b^	0.89 ± 0.02 ^a^	1.85 ± 0.14	91.05 ± 0.65 ^b^	5.98 ± 0.57 ^b^
R	0.76 ± 0.07	4.86 ± 0.40	89.07 ± 0.00 ^a^	568.48 ± 38.03 ^a^	0.63 ± 0.06 ^b^	1.83 ± 0.21	87.67 ± 0.59 ^c^	8.94 ± 0.43 ^a^
EP1	0.75 ± 0.09	4.84 ± 0.34	83.10 ± 0.00 ^b^	450.30 ± 14.90 ^b^	0.64 ± 0.03 ^b^	1.74 ± 0.05	91.24 ± 0.15 ^b^	5.57 ± 0.18 ^b^
EP2	0.82 ± 0.02	4.55 ± 0.31	72.33 ± 0.00 ^c^	410.23 ± 0.00 ^b,c^	0.69 ± 0.03 ^b^	1.59 ± 0.26	92.95 ± 0.14 ^a^	4.61 ± 0.24 ^c^
EP3	0.79 ± 0.08	4.68 ± 0.53	67.48 ± 0.00 ^d^	365.64 ± 12.10 ^c^	0.68 ± 0.06 ^b^	1.71 ± 0.30	93.54 ± 0.28 ^a^	4.35 ± 0.35 ^c^
*F* Value	0.36	0.14	33.14 **	17.02 **	8.42 *	1.07	71.79 **	44.73 **

Notes: C: 30% pork fat; R: 15% pork fat + 10% soybean oil; EP1: R + 1% EP; EP2: R + 2% EP; EP3: R + 3% EP. The data in the table are expressed as means ± standard deviation. ^a–d^: Different letters within the same column mean significant differences (*p* < 0.05), no letters within the same column mean no significant differences (*p* > 0.05). * Symbol represents *p* value (*p* ≥ 0.05, * 0.01 ≤ *p* < 0.05, ** *p* < 0.01).

**Table 6 foods-10-00743-t006:** Texture profile analysis of the pork sausages.

Treatment	Hardness	Springiness	Cohesiveness	Adhesiveness	Chewiness
C	2177.39 ± 100.88 ^b,c^	0.70 ± 0.01	0.68 ± 0.00	1470.24 ± 78.98 ^a,b^	978.36 ± 10.04 ^b,c^
R	1998.12 ± 129.90 ^c^	0.67 ± 0.03	0.63 ± 0.02	1269.82 ± 113.40 ^b^	888.67 ± 62.98 ^c^
EP1	2492.21 ± 104.70 ^a,b^	0.68 ± 0.00	0.67 ± 0.00	1677.04 ± 65.48 ^a,b^	1130.33 ± 44.13 ^a,b^
EP2	2734.66 ± 141.18 ^a^	0.66 ± 0.06	0.64 ± 0.00	1751.48 ± 89.06 ^a,b^	1231.26 ± 27.67 ^a^
EP3	2891.68 ± 167.30 ^a^	0.67 ± 0.03	0.65 ± 0.05	1885.74 ± 264.73 ^a^	1256.62 ± 97.53 ^a^
*F* Value	24.26 **	0.60	1.78	8.67 *	23.56 **

Notes: C: 30% pork fat; R: 15% pork fat + 10% soybean oil; EP1: R + 1% EP; EP2: R + 2% EP; EP3: R + 3% EP. The data in the table are expressed as means ± standard deviation. ^a–c^: Different letters within the same column mean significant differences (*p* < 0.05), no letters within the same column mean no significant differences (*p* > 0.05). * Symbol represents *p* value (*p* ≥ 0.05, * 0.01 ≤ *p* < 0.05, ** *p* < 0.01).

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
