# Peer review of "Effect of Eggplant Powder on the Physicochemical and Sensory Characteristics of Reduced-Fat Pork Sausages"

_foods, 2021, doi:10.3390/foods10040743_

Round 1

Reviewer 1 Report

Referee report on foods-1151937 entitled “Effect of Eggplant Powder on the Physicochemical and Sensory Characteristics of Reduced-fat Pork Sausages”.  

The manuscript foods-1151937 describes the possible effect of the addition of eggplant powder on the physicochemical and sensory properties of reduced-fat pork sausages. I found this study to be novel in that the use of eggplant as an additive in meat products has not been studied before, and due to its natural function, that could improve the physicochemical and sensory properties of reduced fat meat products.

I would also like to see a fatty acid analysis done to see the difference between the SFA, MUFA and PUFA composition, but perhaps this can be done in another paper. The results show that the addition of EP can improve the sensory properties and the water and oil binding properties of reduced fat meat products, which is an important technological parameter in the production of this type of meat products.

I propose some changes that I think are improvements in the whole manuscript. My main concern is to clarify which physicochemical properties have been improved, since the proximate composition has not been improved. So, this has to be corrected especially in abstract and conclusion section.  Also, care should be taken when presenting the results - the results should not be repeated, which is the case with the sensory analysis results.

Necessary changes are listed below.

In the abstract mention what were the percentages of eggplant powder added.

L13 Low-fat pork gel with EP showed remarkably better physicochemical and sensory properties than those without EP- you should be careful with this statement as the proximate composition of the sausages with EP had similar values to those of the R samples, so I suggest correcting and clarifying this in the abstract.

L26 rephrase rich nutrition.

L72 mention the percentage of eggplant powder added in this study.

L119 state manufacturer and country for commercial cholesterol analysis test kit.

L187 but also improved the composition of fatty acids- how can you claim this as no analysis of fatty acid composition was carried out.

L192 the protein content of R and EP1 was lower than that of the other formulations, which may be caused by their poor water- and fat-binding capacities, and the cooking loss contained some water-soluble proteins- you state this, but no statistical difference was found for the protein content in the different formulations, so change this sentence.

L313 the hardness and chewiness values were improved through EP addition- were they really improved, or did the EP addition affect the increase in hardness and chewiness?

Table 7 and Figure 2 show the same results. Do not duplicate the results. I suggest choosing one way to present the sensory analysis results and then correct in the results and discussion section.

L383 physicochemical characteristics were poor and the sensory properties were unacceptable- this is not correct, sensory properties were neutral rather than unacceptable and physicochemical properties of R group were similar to those of group with different percentage of EP.

L389 EP showed potential as a means of enhancing physicochemical characteristics and sensory properties of sausages- improve sensory properties of sausages yes, but physicochemical properties not so much. Proximate composition not so much, so you need to be specific. I propose to correct the conclusion based on the results obtained.

Reviewer 2 Report

Introduction

Please also include a literature review on similar initiatives to this paper, to be specific other papers that attempts to increase nutritional value of (pork) sausages that would be good for the readers to be aware on what is happening as well in the field.

Section 2.4

Which specific AOAC method did they use?

What commercial cholesterol kit? Include brand and country.

Section 2.8

What weight load did the authors use in their machine? 50kg?

Section 2.9

How experienced are the panellist here? Elaborate

mouth-feel is one word, mouthfeel. Correct this

Why did the panellist need to gargle with warm water?

It seems that it is done on a categorical hedonic scale, can the authors provide more reasoning why a hedonic evaluation is done here as to a descriptive sensory profiling? 

What does the author mean by deducting the highest and lowest scores?

How can the authors calibrate the liking of an individual? The approach seems lacking of information and needs further justification.

Results - Tables. Please include the F value as an additional row and indicate the exact p value as to merging them to < .05

Discussion

The authors needs to further dissect and include more information on their references, for example on Section 3.6

What exactly are the positive effects of dietary fiber to sensory properties from previous studies? Expand.

Or Kehlet [45] mentioning balance between sensory and nutrition, what sensory attributes did they measure? 

Conclusion

What are the limitations and future research avenue for this study? One of the pitfalls that the authors did not include design of experiments for example in understanding how different levels of EP can influence the physicochemical and sensory properties of the sausages. One good reference to include would be the use of d-optimal mixture design: https://www.mdpi.com/2304-8158/8/6/214

Generally, more information is needed in this manuscript. 

Round 2

Reviewer 2 Report

I'd like to thank the authors for their response to the review. However there are comments that hasn't been addressed.

In regards to weight, I'm meaning the loadcell that the authors used not the trigger force - see here: https://www.stablemicrosystems.com/TextureAnalysisAccessories.html

Another question then raises why did the authors measure liking with a sensory panel than consumers? This needs to be added as a limitation in their study.

In terms of the Grubbs test, please revise abnormal values to outliers.
